# Robust Control with Uncertain Disturbances for Vehicle Drift Motions

Dongxin Xu 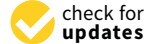, Guoye Wang *, Longtao Qu and Chang Ge

College of Engineering, China Agricultural University, Beijing 100083, China; xudongxin1996@163.com (D.X.); 15839191315@163.com (L.Q.); 18811738966@163.com (C.G.)
* Correspondence: wgy1615@126.com; Tel.: +86-010-6273-6856

**Abstract:** Professor drivers, including racing drivers, can drive cars to achieve drift motions by taking control of the steering angle in high tire slip ratios, which provides a way to improve the driving safety of autonomous vehicles. The existing studies can be divided into two kinds based on analysis methods, and the theory-based is chosen in this study. Because the recent theory based is most applied for planar models with neglect of the rollover accident risk, the nonlinear vehicle model is established by considering longitudinal, lateral, roll, and yaw motions and rolling safety with the nonlinear tire model UniTire. The drift motion mechanism is analyzed in steady and transient states to obtain drift motion conditions, including the velocity limitation and the relationship between sideslip angle and yaw rate, and vehicle main status parameters including the velocity, side-slip angle and yaw rate in drift conditions. The state-feedback controller is designed based on robust theory and LMI (linear matrix inequation) with uncertain disturbances to realize circle motions in drift conditions. The designed controller in simulations realizes drift circle motions aiming at analyzed status target values by matching the front-wheel steering angle with saturated tire forces, which satisfies the Lyapunov stability with robustness. Robust control in drift conditions solves the problem of how to control vehicles to perform drift motions with uncertain disturbances and improves the driving safety of autonomous vehicles.

**Keywords:** vehicle drift motion; motion mechanism analysis; robust control



## 1. Introduction

Professor drivers, including racing drivers, can drive cars to accomplish drifts and sharp turns in drift conditions, which means tire forces reach the maximum and tires are in high slip ratios and is very dangerous for ordinary drivers but can ensure maximum safety. The autonomous vehicle will be safer if it can realize drift motions. The paper studies how to control vehicles in drift motions.

The studies of vehicle drift control are divided into two kinds based on analysis methods, one of which is experience-based and the other is theory-based. The experience-based is designed by artificial intelligence, including neural networks, based on the professor drivers' data. Cutler designed a PILCO (probabilistic inference for learning control) controller based on the combination of simulators and a real-world robot car to realize a steady-state drift motion by learning the optimal solutions of a simple model in [1]. Acosta proposed a hybrid structure formed by the MPC (model predictive controller) and NNs (neural networks) to achieve drift motions in which the NNs provided the motion references and tire parameters by learning in [2]. Spielberg applied a feedforward-feedback controller based on the neural network to realize vehicle drift motions by learning experimental and simulated data in [3]. Cai designed a controller based on the model-free deep reinforcement learning algorithm soft actor-critic to realize vehicle drift motions after being trained on tracks in [4]. Artificial intelligence is most often expressed in stochastic algorithms that often have no knowledge whatsoever of the underlying problem being learned



(a considerable strength of the methods) in [5]. The theory-base is designed by control theories based on drift motion analyses. The drift circle motion is the first and important step of the vehicle drift study. Velenis obtained the main state parameters including the vehicle velocity and the sideslip angle and yaw rate by analyzing the steady-state cornering based on a three-state model in drift conditions in [6]. Hindiyeh analyzed the drift equilibrium of the vehicle phase portrait based on the three-state bicycle model, presented a controller framework for autonomous drifting of a rear-wheel-drive vehicle, and achieved the drift motion in [7,8]. Bobier-Tiu analyzed vehicle stability properties and equilibrium point locations and movement to changing parameters and system inputs by phase portraits based on the bicycle model and applied phase portrait analysis to the controller design in drift motions in [9]. Huang analyzed the vehicle equilibrium and designed an equilibrium condition calculator based on the pre-distribution of the longitudinal force of the rear tire of a four-wheel vehicle model in drift conditions in [10]. Park analyzed drift equilibrium states based on a three-state bicycle model and obtained the equilibrium status values of the body slip angle and yaw rate with low velocities when the vehicle rotated clockwise and counter-clockwise in [11]. Each of the two kinds of studies has its advantages and disadvantages. The experience-based option can be more human to achieve the drift motions than the theory-based one, but the theory-based option can have no need for operational data, which is different for various combinations of vehicles and roads and is difficult to obtain for most studies. Besides, the theory-based studies are mainly applied to vehicle planar models with the neglect of the large rolling motion which in drift conditions is possible regarding the vehicle behaviors controlled by drivers, and there are high risks of rollover with the large rolling. Therefore, this paper studies the vehicle drift control based on the drift motion analyses in consideration of rolling safety.

Choosing an appropriate algorithm of the controller also is significant to achieve drift motions, which can be decided by the references of the existing vehicle drift control methods and vehicle path tracking methods. Most theory-based controllers are designed based on the optimizing control theory. Velenis in [6], Huang in [10], and Park in [11] all designed controllers based on the LQR (linear quadratic regulator) theory. Wachter designed an optimal controller based on the SDRE (state dependent Riccati equation) technique and implemented the controller in a test vehicle in [12]. Bardos designed a MIMO (multiple input multiple output) LQR controller by a three-state bicycle model with saturated rear tires to realize vehicle drift motions in steady states in [13,14]. The MPC theory also is an optimizing control theory and has a great ability to handle linear constraints and future prediction in the design process, which is popular in path tracking as shown by Tan in [15] and Bai in [16]. The optimizing controllers can obtain the optimum solutions based on accurate models without disturbances, and the anti-jamming capability of optimizing controllers is worse than robust controllers which direct at uncertain problems. The robust theory, in recent years, is used to design vehicle controllers to follow paths and track, which provides a reference for the controller design. Boyali designed controllers by LQ (linear quadratic) H-infinity robust in LMI (linear matrix inequation) using the LPV (linear parameter varying) vehicle models to achieve path tracking in [17]. Li designed an H-infinity controller with an uncertainty model and shown the effectiveness by simulation experiments in [18]. He designed a robust H-infinity coordination controller based on LMI with the consideration of uncertain external disturbance to realize vehicle path tracking in [19]. Zhang designed an H-infinity controller with a T-S (Takagi-Sugeno) fuzzy model to control the front-wheel steering for path tracking with parametric uncertainties and nonlinearities in [20]. The H-infinity control is one kind of robust control and is suitable to deal with vehicle modeling uncertainty and external disturbance as [21] mentioned. Considering that there are uncertain external disturbances in practice, the paper designs the drift controller based on the H-infinity control.

From the above analysis, the paper studies how the vehicle work in drift conditions by analyzing the vehicle motion mechanism with the reference of the human operation. The vehicle dynamics model is established by considering longitudinal, lateral, roll, and

yaw motions and the rolling safety with the nonlinear tire model UniTire. The drift motion mechanism is analyzed in steady and transient states to obtain vehicle main status parameters. The paper designs the drift controller based on the H-infinity and LMI with uncertain external disturbances and the controller is proved by simulations.

## 2. Vehicle Drift Motion Mechanism Analysis

The motion mechanism analysis is primary for the whole, which suggests how the vehicle moves in drift conditions. This section analyzes the vehicle roll safety and motion mechanisms in drift conditions, based on an appropriate dynamics model which is enough to describe vehicle motion characteristics without redundancy.

### 2.1. Vehicle Dynamics Model

The mechanism analysis and the controller design are based on the 4-DOF nonlinear vehicle dynamics model depicted in Figure 1. The model is established with the nonlinear tire characteristics and the normal load transfer from front to rear wheels arising from the longitudinal acceleration of the vehicle to describe the motion in drift conditions.

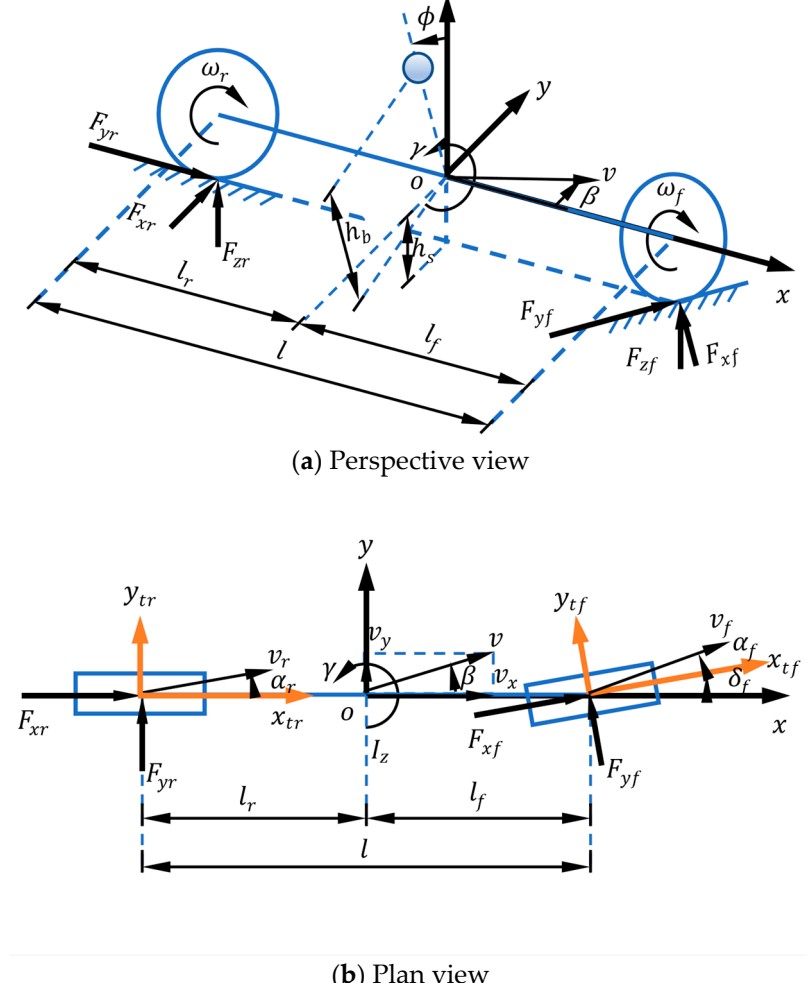

(**a**) Perspective view

(**b**) Plan view

**Figure 1.** Vehicle dynamics model.

2.1.1. Vehicle Dynamics Model Equations

In order to ensure driving safety, vehicle wheels remain in contact with the ground under drift conditions. A 4-DOF nonlinear vehicle dynamics model is established to describe drift motion in Figure 1. The *o-x-y-z* coordinate system is used to describe the chassis coordinate system, where the *o*-point is the origin of the coordinate system, the horizontal

$x$-axis is in vehicle longitudinal motion direction, the horizontal $y$-axis is perpendicular to the $x$-axis and the vertical $z$-axis is positive in the upward direction. The $x_{tf}$-$y_{tf}$ and the $x_{tr}$-$y_{tr}$ coordinate systems are used to describe the horizontal coordinate systems of the front and rear tires, where the $x_{tf}$-axis and the $x_{tr}$-axis are in tire revolution directions. Considering vehicle dynamics characteristics, the model is established with longitudinal, lateral, roll, and yaw motions, and the equations of motion are as shown in Equation (1).

$$\begin{cases} m\dot{v}_x - mv_y\gamma + m_b h_b \dot{\phi}\gamma = F_{xf}\cos\delta_f - F_{yf}\sin\delta_f + F_{xr} - F_d \\ m\dot{v}_y + mv_x\gamma - m_b h_b \ddot{\phi} = F_{xf}\sin\delta_f + F_{yf}\cos\delta_f + F_{yr} \\ I_x\ddot{\phi} - I_{xz}\dot{\gamma} - m_b h_b\left(\dot{v}_y + v_x\gamma\right) = m_b g h_b \sin\phi - K_\phi\phi - C_\phi\dot{\phi} \\ I_z\dot{\gamma} - I_{xz}\ddot{\phi} = l_f\left(F_{xf}\sin\delta_f + F_{yf}\cos\delta_f\right) - l_r F_{yr} \end{cases} \tag{1}$$

where $v_x = v\cos\beta$, $v_y = v\sin\beta$, $F_d = 0.5\rho_a C_d A_f v_x^2$.

Considering load transfer caused by lateral force, the equilibrium of forces in the vertical direction and the equilibrium of moments are used to find front and rear axle normal loads:

$$\begin{cases} F_{zf} + F_{zr} - mg = 0 \\ l_f F_{zf} - l_r F_{zr} + \sum F_x h_g = 0 \end{cases} \tag{2}$$

where $\sum F_x = m\dot{v}_x - mv_y\gamma + m_b h_b \dot{\phi}\gamma = F_{xf}\cos\delta_f - F_{yf}\sin\delta_f + F_{xr} - F_d$.

### 2.1.2. Tire Force

The UniTire model is a nonlinear and non-steady-state tire model for vehicle dynamics simulation and control to describe tire properties accurately under complex conditions involving the large lateral slip, the large longitudinal slip and the camber in [22–25], which can be used to describe tire properties in the studied conditions. In this section, some inferences based on the tire model are described to simplify the drift motion mechanism analysis and the whole equations of UniTire are not introduced.

The longitudinal and lateral slip ratios at each tire are defined as:

$$S_{xi} = \frac{\omega_i r_{ei} - v_{xi}}{\omega_i r_{ei}}, \ S_{yi} = -\frac{v_{yi}}{\omega_i r_{ei}} = (S_{xi} - 1)\tan\alpha_i$$

where $\tan\alpha_i = v_{yi}/v_{xi}$. The corresponding velocities along the wheel's longitudinal and lateral axes of the front and rear wheels are given by:

$$v_{xf} = v_x\cos\delta_f + \left(v_y + l_f\gamma\right)\sin\delta_f, \ v_{xr} = v_x v_{yf} = -v_x\sin\delta_f + \left(v_y + l_f\gamma\right)\cos\delta_f, \ v_{yr} = v_y - l_r\gamma$$

The normalized longitudinal, lateral, and combined slip ratios at each tire are defined as:

$$\phi_{xi} = \frac{K_{xi}S_{xi}}{\mu_{xi}F_{zi}}, \ \phi_{yi} = \frac{K_{yi}S_{yi}}{\mu_{yi}F_{zi}}, \ \phi_i = \sqrt{\phi_{xi}^2 + \phi_{yi}^2}$$

The normalized longitudinal and lateral forces on each tire are described as:

$$\overline{F_{xi}} = \frac{F_{xi}}{\mu_{xi}F_{zi}} = \overline{F_i}\frac{\phi_{xi}}{\phi_i}, \ \overline{F_{yi}} = \frac{F_{yi}}{\mu_{yi}F_{zi}} = \overline{F_i}\frac{\phi_{yi}}{\phi_i}$$

The utilization of friction coefficient at each tire stays the same at the ultimate value in drift conditions, so that force on each tire reaches its maximum. Combined with UniTire, the longitudinal, lateral and resultant forces on each tire are derived as:

$$F_{xi} = \frac{K_{xi}S_{xi}}{K_{yi}S_{yi}}F_{yi}, \ F_i = \sqrt{F_{xi}^2 + F_{yi}^2} = \mu_i F_{zi} \tag{3}$$

### 2.1.3. Roll Safety Analysis

Vehicle wheels lift off from the ground more often than not as a result of large rolling motion under drift conditions. The vehicle has a risk of rollover when the vertical force on one side wheel equals zero, otherwise, there is no risk in [26–28]. Therefore, the vehicle roll model is classified into two conditions: before and after wheel lift-off as shown in Figure 2a,b.

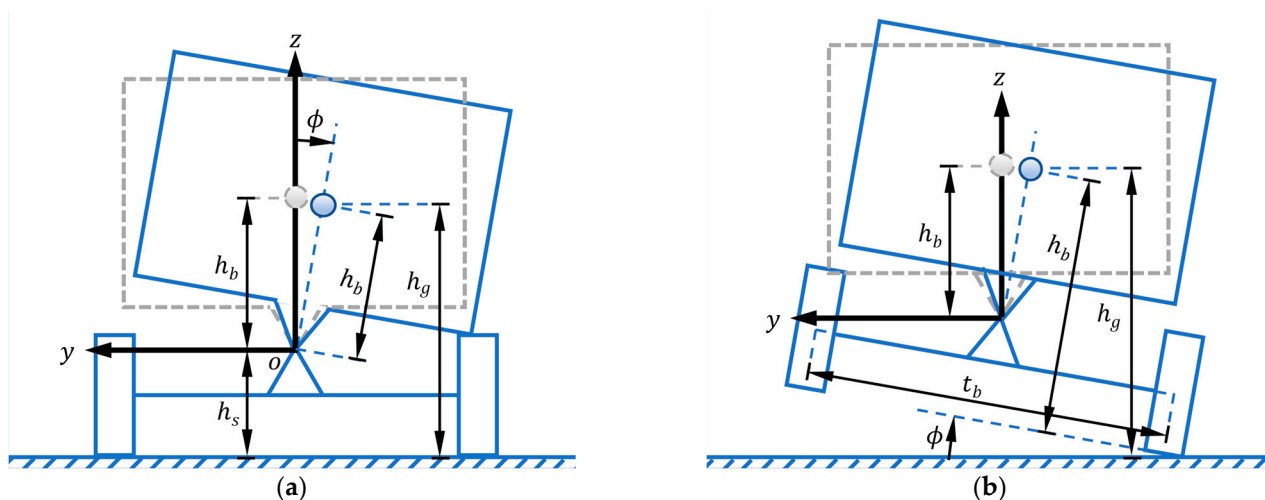

**Figure 2.** Roll dynamic model should be listed as: (**a**) before wheel lift-off; (**b**) after wheel lift-off.

The roll dynamics of the vehicle is represented by two different equations as expressed in Equations (4) and (5) corresponding to Figure 2a,b:

$$I_x\ddot{\phi} - I_{xz}\dot{\gamma} - m_b h_b\left(\dot{v}_y + v_x\gamma\right) = m_b g h_b \sin\phi - K_\phi\phi - C_\phi\dot{\phi} \tag{4}$$

$$I_{x\phi}\ddot{\phi} - I_{xz\phi}\dot{\gamma} - m h_b\left(\dot{v}_y + v_x\gamma\right) = m g h_b \sin\phi - m g \frac{t_b}{2}\cos\phi \tag{5}$$

Apparently, the relational expression between the safe roll angle $\phi_s$, the safe roll rate $\dot{\phi}_s$ and the safe roll angular acceleration $\ddot{\phi}_s$ in critical states can be obtained.

### 2.2. Motion Mechanism Analysis in Steady States

The vehicle motion in the steady state is in a straight line or in a constant circle with a constant velocity, while the vehicle status parameters including the sideslip angle, the yaw rate and the roll angle also are constant:

$$v = v^s, \ \gamma = \gamma^s = \frac{v^{s2}}{v_x^s R^s} = \frac{v^s}{R^s \cos\beta^s}, \ \beta = \beta^s, \ \phi = \phi^s$$

Then the derivative parameters of the above, including vehicle accelerated velocity, equal zero:

$$\dot{v} = 0, \ \dot{\beta} = 0, \dot{\gamma} = 0, \ \ddot{\phi} = \dot{\phi} = 0$$

It is clear that vehicle dynamics model equations in steady states are derived as:

$$0 = F_{xf}{}^s \cos\delta_f{}^s - F_{yf}{}^s \sin\delta_f{}^s + F_{xr}{}^s - F_d{}^s + mv^s\gamma^s \sin\beta^s \tag{6}$$

$$0 = F_{xf}{}^s \sin\delta_f{}^s + F_{yf}{}^s \cos\delta_f{}^s + F_{yr}{}^s - mv^s\gamma^s \cos\beta^s \tag{7}$$

$$0 = m_b g h_b \sin\phi^s - K_\phi\phi^s + m_b h_b v^s\gamma^s \cos\beta^s \tag{8}$$

$$0 = l_f\left(F_{xf}{}^s \sin\delta_f{}^s + F_{yf}{}^s \cos\delta_f\right) - l_r F_{yr}{}^s \tag{9}$$

In the following, the rear and front wheel slip ratios in drift conditions are derived, which needs to satisfy in order for the vehicle to maintain a steady-state condition.

### 2.2.1. Rear Axle Steady-State Equations

Combining Equations (7) and (9), the steady-state lateral force at the rear wheel $F_{yr}{}^s$ is derived as:

$$F_{yr}{}^s = \frac{l_f}{l_f + l_r} mv^s \gamma^s \cos \beta^s \tag{10}$$

Combining Equation (2) and constant parameters, the steady-state vertical forces $F_{zf}{}^s$ and $F_{zr}{}^s$ are derived as:

$$F_{zf}{}^s = \frac{l_r}{l_f + l_r} mg + \frac{h_g}{l_f + l_r} mv^s \gamma^s \sin \beta^s \tag{11}$$

$$F_{zr}{}^s = \frac{l_f}{l_f + l_r} mg - \frac{h_g}{l_f + l_r} mv^s \gamma^s \sin \beta^s \tag{12}$$

Combining Equations (3), (10) and (12), the forces at the rear wheel can be obtained in steady-state drift conditions.

### 2.2.2. Front Axle Steady-State Equations

Combining Equations (6) and (7), the steady-state resultant force at the front wheel $F_f{}^s$ is derived as:

$$F_f{}^s = \sqrt{\left(F_{xr}{}^s - F_d{}^s + mv^s \gamma^s \sin \beta^s\right)^2 + \left(F_{yr}{}^s - mv^s \gamma^s \cos \beta^s\right)^2} \tag{13}$$

The normalized longitudinal slip ratio at the front tire is relatively small in drift conditions, so that Equation (3) is derived as:

$$F_{xf} \approx 0, \ F_{yf} \approx F_f = \mu_f F_{zf} \tag{14}$$

Combining Equations (11), (13) and (14), the forces at the front wheel can be obtained in steady-state drift conditions.

### 2.3. Motion Mechanism Analysis in Transient States

The vehicle has two motion states in practice, where one is the steady state mentioned and the other is the transient state between the current steady state and the targeted steady state both of which satisfy the motion mechanism in Section 2.2. The vehicle moves in a circle with a variable radius, while the vehicle longitudinal velocity, the lateral velocity, the sideslip angle, the yaw rate and the roll angle in transient states are as follows:

$$\gamma = \gamma^t, \ \phi = \phi^t, \beta = \beta^t, \ v = v^t R = R^t = \frac{v^2}{v_x \gamma + \dot{v}_y} = \frac{v^{t2}}{v^t \gamma^t \cos \beta^t + \dot{v} \sin \beta^t + v^t \dot{\beta} \cos \beta^t}$$

It is assumed that it takes one-unit time from the current state to the target state, so differences of parameters in states can express the derivatives. The derivative parameters of the above, including vehicle accelerated velocity, are derived as:

$$\dot{v} = \Delta v, \ \dot{\beta} = \Delta \beta, \ \dot{\gamma} = \Delta \gamma, \ \dot{\phi} = \Delta \phi, \ \ddot{\phi} = \Delta \dot{\phi}$$

Obviously, the vehicle dynamics model equations in transient states are derived as:

$$m\left(\Delta v \cos \beta^t - v^t \Delta \beta \sin \beta^t\right) - mv^t \gamma^t \sin \beta^t + m_b h_b \gamma^t \Delta \phi = F_{xf}{}^t \cos \delta_f{}^t - F_{yf}{}^t \sin \delta_f{}^t + F_{xr}{}^t - F_d{}^t \tag{15}$$

$$m\left(\Delta v \sin \beta^t + v^t \Delta \beta \cos \beta^t\right) + mv^t \gamma^t \cos \beta^t - m_b h_b \Delta \dot{\phi} = F_{xf}{}^t \sin \delta_f{}^t + F_{yf}{}^t \cos \delta_f{}^t + F_{yr}{}^t \tag{16}$$

$$I_x \Delta \dot{\phi} - I_{xz} \Delta \gamma - m_b h_b (\Delta v \sin \beta^t + v^t \Delta \beta \cos \beta^t + v^t \gamma^t \cos \beta^t) = m_b g h_b \sin \phi^t - K_\phi \phi^t + C_\phi \Delta \phi \tag{17}$$

$$I_z \Delta \gamma - I_{xz} \Delta \dot{\phi} = l_f \left( F_{xf}{}^t \sin \delta_f{}^t + F_{yf}{}^t \cos \delta_f{}^t \right) - l_r F_{yr}{}^t, \tag{18}$$

The roll angular acceleration needs to satisfied Equation (17), and the rear and front wheel slip ratios in drift conditions are derived in the following, which needs to satisfy in order for the vehicle to maintain a transient-state condition.

### 2.3.1. Rear Axle Transient-State Equations

Combining Equations (16) and (18) in transient states, the transient-state lateral force at the rear wheel $F_{yr}{}^t$ is derived as:

$$F_{yr}{}^t = \frac{l_f}{l_f + l_r} m \left( v^t \gamma^t \cos \beta^t + \Delta v \sin \beta^t + v^t \Delta \beta \cos \beta^t - \frac{m_b}{m} h_b \Delta \dot{\phi} \right) - \frac{I_z}{l_f + l_r} \Delta \gamma + \frac{I_{xz}}{l_f + l_r} \Delta \dot{\phi} \tag{19}$$

Combining Equation (2) and variable parameters, the transient-state vertical forces $F_{yf}{}^t$ and $F_{zr}{}^t$ are derived as:

$$F_{zr}{}^t = \frac{l_f}{l_f + l_r} mg + \frac{h_g}{l_f + l_r} m \left( \Delta v \cos \beta^t - v^t \Delta \beta \sin \beta^t - v^t \gamma^t \sin \beta^t \right) + \frac{h_g}{l_f + l_r} m_b h_b \gamma^t \Delta \phi \tag{20}$$

$$F_{zf}{}^t = \frac{l_r}{l_f + l_r} mg - \frac{h_g}{l_f + l_r} m \left( \Delta v \cos \beta^t - v^t \Delta \beta \sin \beta^t - v^t \gamma^t \cos \beta \right) - \frac{h_g}{l_f + l_r} m_b h_b \gamma^t \Delta \phi \tag{21}$$

Combining Equations (3), (19) and (20), the forces at the rear wheel can be obtained in transient-state drift conditions.

### 2.3.2. Front Axle Transient-State Equations

Combining Equations (15) and (16), the transient-state resultant force at front wheel $F_f{}^t$ is derived as:

$$F_f{}^t = \sqrt{ \begin{array}{l} \left( F_{xr}{}^t - F_d{}^t + mv^t \gamma^t \sin \beta^t - m\Delta v \cos \beta^t + mv^t \Delta \beta \sin \beta^t - m_b h_b \gamma^t \Delta \phi \right)^2 \\ + \left( F_{yr}{}^t + m_b h_b \Delta \dot{\phi} - mv^t \gamma^t \cos \beta^t - m\Delta v \sin \beta^t - mv^t \Delta \beta \cos \beta^t \right)^2 \end{array} } \tag{22}$$

Combining Equations (14), (21) and (22), the forces at the front wheel can be obtained in transient-state drift conditions.

## 3. Robust Control in Drift Conditions

This section designs the controller based on robust with uncertain external disturbances to realize drift motions. The control system is described as state space representation with the linearized tire model. The whole equations of UniTire are used in the simulation model to sufficiently describe tire characteristics and the tire model is linearized in the control system to simplify the controller state-space expressions.

The linearized tire model and the affine expression can optimize the controller design in [29]. According to UniTire, the tire longitudinal and lateral forces are related to the tire slip ratios, the tire slip angle, the wheel center velocity, and the tire vertical force. The tire longitudinal slip ratio has a greater influence on the tire longitudinal force and the tire slip angle affects the tire lateral force relatively large. The affine functions of the tire longitudinal and lateral forces are shown as Equation (23) in Figure 3.

$$\begin{cases} F_{xi} \approx \widetilde{F}_{xi} = \overline{C}_{xi} (\kappa_i - \overline{\kappa}_i) + \overline{F}_{xi} \\ F_{yi} \approx \widetilde{F}_{yi} = \overline{C}_{yi} (\alpha_i - \overline{\alpha}_i) + \overline{F}_{yi} \end{cases} \tag{23}$$

where $\kappa_i = -(v_{xi} - \omega_i r_{ei})/v_{xi} = S_{xi}/(S_{xi} - 1)$ is the TYDEX longitudinal slip ratio, and $\overline{\kappa}_i, \overline{\alpha}_i, \overline{C}_{xi}, \overline{C}_{yi}, \overline{F}_{xi}$, and $\overline{F}_{yi}$ are the known slip ratio, slip angle, slip stiffness, cornering stiffness, longitudinal force, and lateral force of the approximation point respectively.

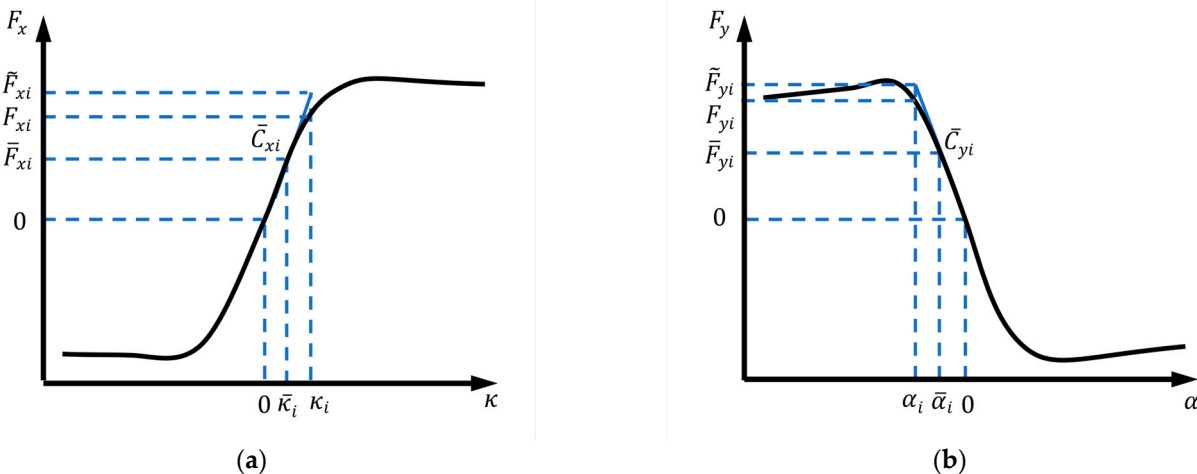

**Figure 3.** UniTire tire model with affine approximation: (**a**) longitudinal force; (**b**) lateral force.

Combining Equations (1) and (23) and the drift motion mechanism analysis, the vehicle state space representation is $\dot{x} = Ax + Bu$. Considered with uncertain external disturbances a control system can be described as:

$$\begin{cases} \dot{x} = Ax + Bu + d \\ \quad y = x \end{cases} \tag{24}$$

where $x = [v - v^e, \beta - \beta^e, \gamma - \gamma^e, \phi - \phi^e]^T$, $u = [\delta_f - \delta_f{}^e, \kappa_f - \kappa_f{}^e, \kappa_r - \kappa_r{}^e]^T$ and $y$ are the state, input and output vectors of the system respectively, and the coefficient matrices $A$ and $B$ are the Jacobian matrices with respect to the state and input vectors. The uncertain external disturbance $d$ is defined as:

$$d = Bd_e \tag{25}$$

where $d_e = \left[d_\delta, d_f, d_r\right]^T$, and $d_\delta$, $d_f$ and $d_r$ are uncertain external disturbances related to three inputs respectively.

The robust control law is designed as:

$$u = Kx \tag{26}$$

where $K$ is the control gain matrix.

The cost function is shown as:

$$\frac{\int_0^t y^T y \mathrm{d}t}{\int_0^t d_e{}^T d_e \mathrm{d}t} \leq \rho^2 \tag{27}$$

where $\rho$ is the scalar to express the anti-disturbance capability and $\rho > 0$.

Supposing that the control system is internally stable, the Lyapunov stability is satisfied and the Lyapunov function and the derived function are selected as:

$$L = x^T P x \tag{28}$$

$$\dot{L} = \dot{x}^T P x + x^T P \dot{x} \tag{29}$$

where $P$ is the positive definite symmetric matrix.

To obtain the gain matrix under the Lyapunov stability, the inequation Equation (30) needs to be satisfied and Equation (31) is established based on Equations (24)–(26), (29) and the two definitions $\lambda = \left[x^T \ d_e{}^T\right]^T$ and $Q = P(A + BK)$:

$$\dot{L} + y^T y - \rho^2 d_e{}^T d_e = \lambda^T \left[ \begin{array}{cc} Q^T + Q + E & PB \\ B^T P & -\rho^2 \cdot E \end{array} \right] \lambda \le 0 \tag{30}$$

$$\left[ \begin{array}{cc} Q^T + Q + E & PB \\ B^T P & -\rho^2 \cdot E \end{array} \right] < 0 \tag{31}$$

Integrating Equation (29) and combining Equation (30), the following inequation is derived:

$$\|x\|^2 \le \frac{\int_0^t \rho^2 d_e{}^T d_e \mathrm{d}t + L(0)}{P_{\min}} \tag{32}$$

Therefore, the system is stable and bounded when the time approaches infinity, which means the system is convergent under external disturbances.

Based on the Schur complement theorem in [19,30,31] and $Q = P(A + BK)$, Equation (31) can be transformed into:

$$\left[ \begin{array}{ccc} A^T P + K^T B^T P + PA + PBK & PB & E \\ B^T P & -\rho^2 \cdot E & 0 \\ E & 0 & -E \end{array} \right] < 0 \tag{33}$$

There are two unknown variable matrices $P$ and $K$ combined in the nonlinear form, so it's difficult to obtain the two matrices by direct solutions. The variable substitution is used to transform Equation (33) as the equivalent inequation to obtain the two matrices. Multiplied the matrix $diag\{P^{-1}\, E\, E\}$ by both sides of the above, the system solution can be transformed as the solution of the following LMIs by defining the two matrices $R_1 = P^{-1}$ and $R_2 = KP^{-1}$:

$$\begin{array}{c} R_1 > 0 \\ \left[ \begin{array}{ccc} R_1 A^T + R_2{}^T B^T + AR_1 + BR_2 & B & R_1 \\ B^T & -\rho^2 \cdot E & 0 \\ R_1 & 0 & -E \end{array} \right] < 0 \end{array} \tag{34}$$

The system gain matrix of the robust controller can be obtained to realize the vehicle drift motion by the solution of Equation (34).

## 4. Simulation and Discussion

In this section, the driving performance of the drift motion is described by analyzing the mechanism result and the satisfying performance of the robust Controller is verified with uncertain disturbances in MATLAB/Simulink.

### 4.1. Motion Mechanism Analysis Result

The vehicle main parameters are shown as Table 1.

**Table 1.** The vehicle main parameters.

| Parameter | Unit | Value |
|:---:|:---:|:---:|
| $m$ (vehicle mass) | kg | 1126.7 |
| $m_b$ (vehicle body mass) | kg | 1111.0 |
| $l_f$ (distance from gravity center to front axle) | m | 1.265 |
| $l_r$ (distance from gravity center to rear axle) | m | 1.335 |
| $h_g$ (height of gravity center) | m | 0.518 |
| $\mu$ (friction coefficient) | | 0.7 |

The steady-state motion mechanism of the vehicle is analyzed as Section 2.2, and the transient-state motion mechanism is directly used as a target in the controller. Considering the maximum steering angle is 0.7 rad in practice, variation trends of the sideslip angle and

yaw rate and the velocity limitation in drift conditions are shown in Figure 4 by analyzing counterclockwise circle motions with radiuses from 2 m to 18 m.

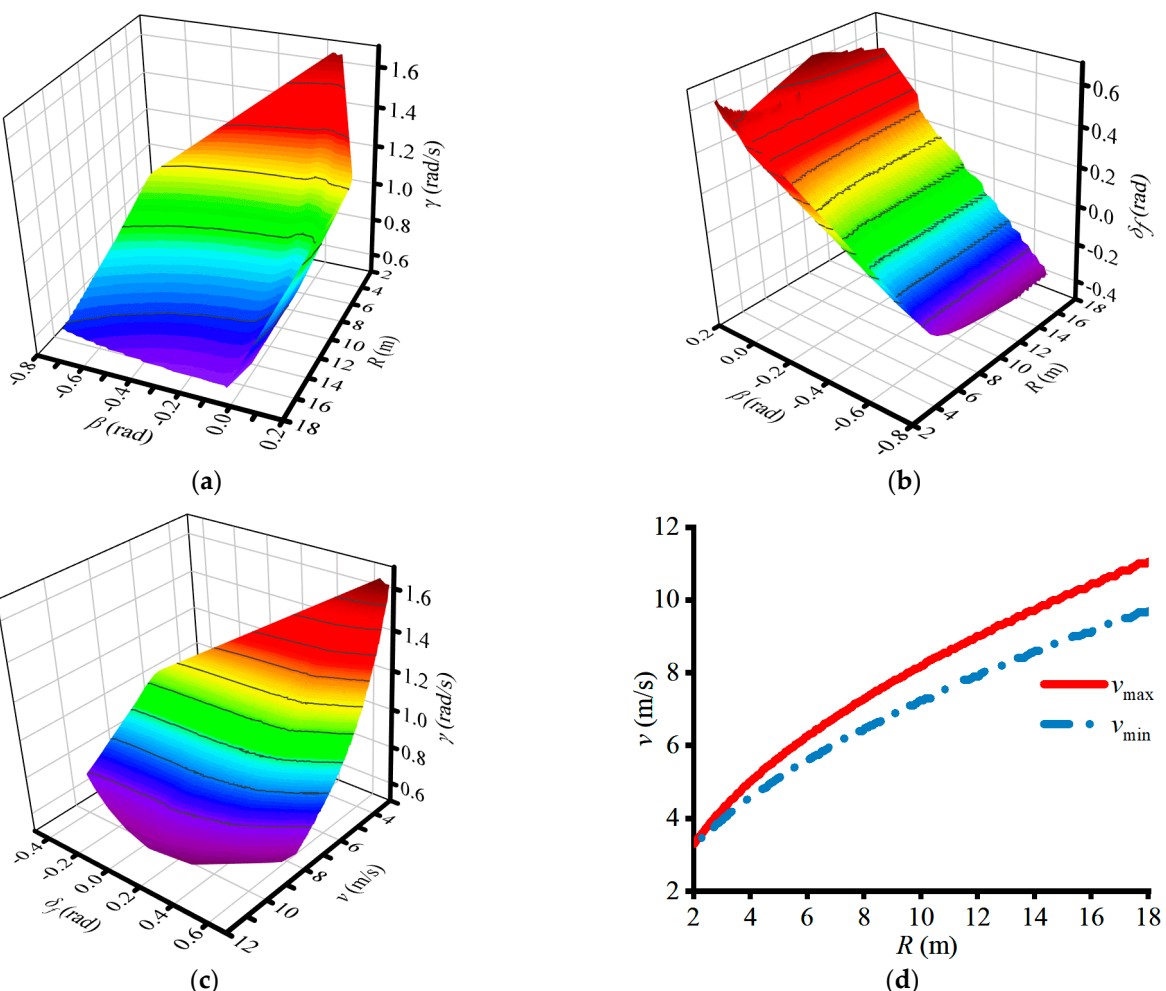

**Figure 4.** The motion mechanism analysis result in drift conditions: (**a**) The relation between the sideslip angle, the yaw rate and the radius; (**b**) the relation between the sideslip angle, the steering angle and the radius; (**c**) the relation between the steering angle, the yaw rate and the velocity; and (**d**) the velocity limitation.

Because circle motions are counterclockwise, according the equation of $\gamma$ in Section 2.2, values of yaw rates are greater than zero. Radiuses in Figures 4–6 all are obtained based on the equation of $R$ in Section 2.3. According to Figure 4a, it suggests that the yaw rate is basically smaller with the larger radius, which means the relationship substantially conforms the negative correlation; the minimum yaw rate consists in the maximum circle with the 18 m radius and equals 0.5 rad/s, which is larger than the circle motion in non-limit conditions. According to Figure 4b, it suggests that the sideslip angle is basically larger with the larger steering angle, which means the relationship substantially conforms the positive correlation; combined with Figure 4a, the sideslip angle is not associated with the radius. According to Figure 4c, it suggests that the yaw rate is basically smaller with the larger velocity, which means the relationship substantially conforms the negative correlation; and the steering angle is not associated with the velocity. According to Figure 4d, it suggests that the velocity is positively associated with the radius, and the maximum and minimum permissible limit of the vehicle velocity $v$ in drift motions are obtained.

**Table 2.** The vehicle main status parameters in drift motions based on the steady-state analysis.

|  | $R$ (m) | $v$ (m/s) | $\beta$ (rad) | $\gamma$ (rad/s) | $\delta_f$ (rad) | $\kappa_r$ |
|---|---|---|---|---|---|---|
| (a) | 16 | 9.5 | −0.56 | 0.7 | −0.29 | 0.72 |
| (b) [1] | 9.6 | 8 | −0.08 | 0.84 | 0.13 | 0.07 |

[1] Group (b) is one part of Figure 4d with the maximum velocity at the radius 9.6 m.

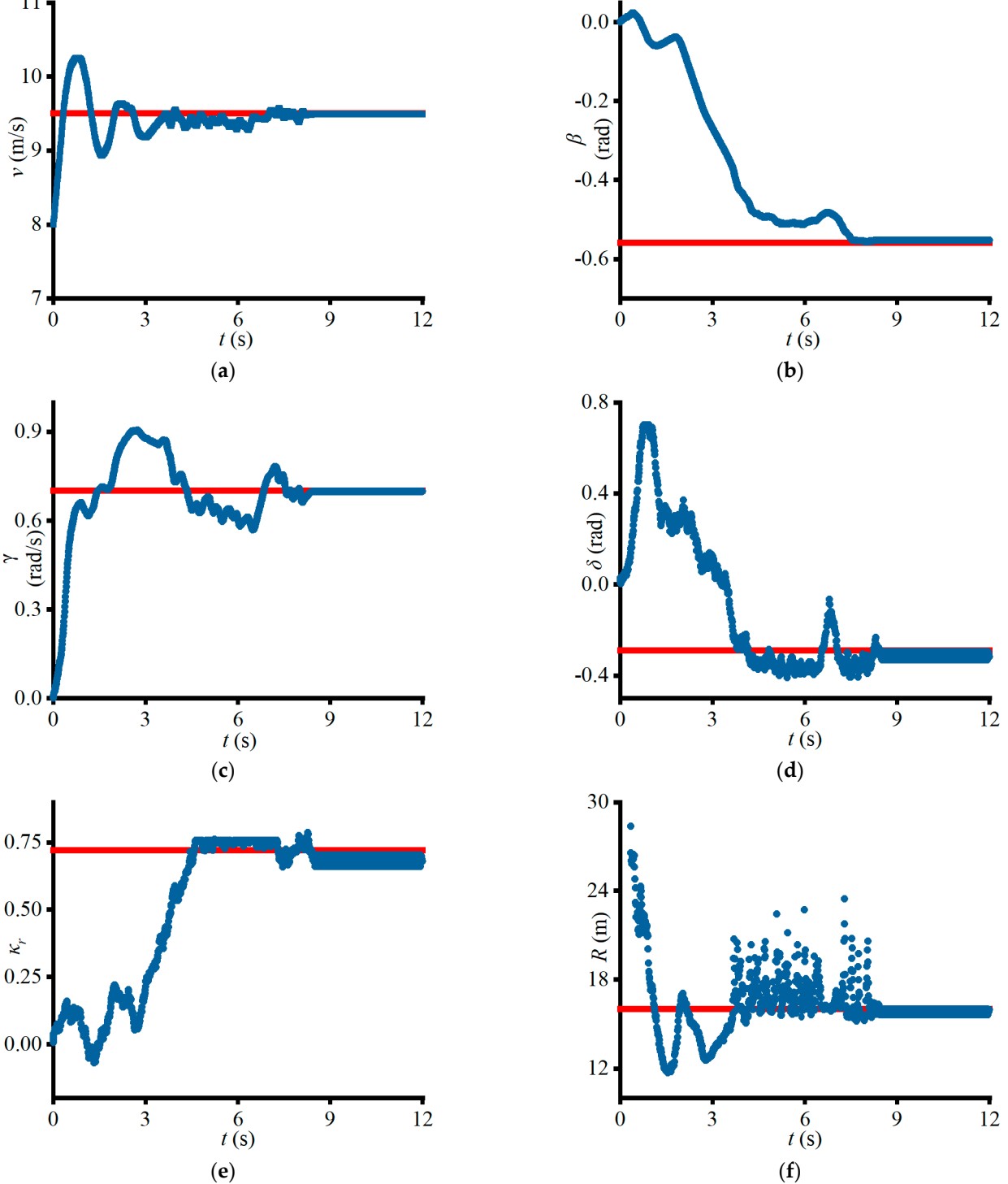

**Figure 5.** The simulation result of group (a) in Table 2 with the origin uniform linear: (**a**) The variation curve of the velocity; (**b**) the variation cure of the sideslip angle; (**c**) the variation curve of the yaw rate; (**d**) the variation cure of the steering angle; (**e**) the variation curve of the TYDEX longitudinal slip ratio of the rear tire; and (**f**) the variation cure of the radius.

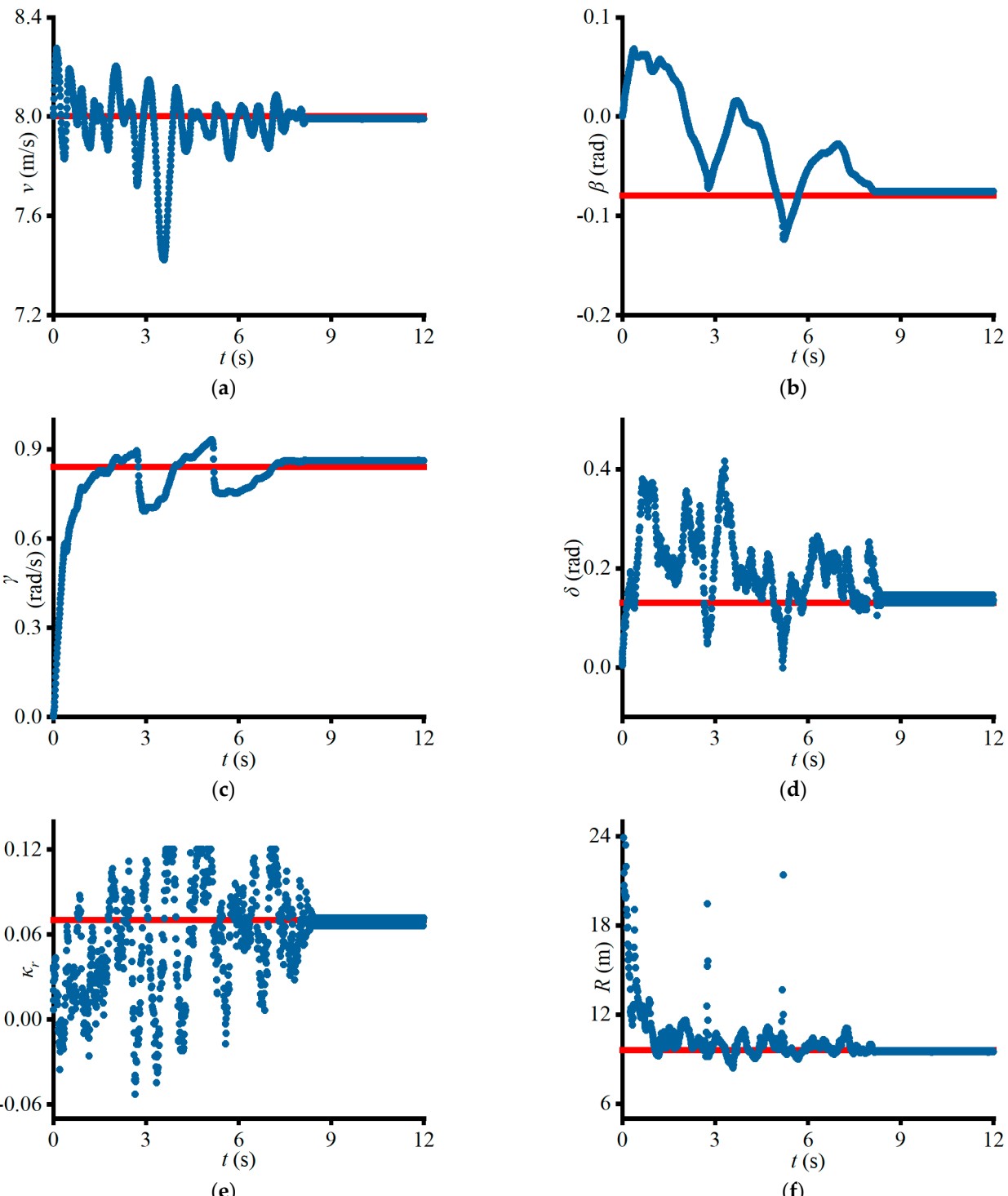

**Figure 6.** The simulation result of group (b) in Table 2 with the origin uniform linear: (**a**) the variation curve of the velocity; (**b**) the variation curve of the sideslip angle; (**c**) the variation curve of the yaw rate; (**d**) the variation curve of the steering angle; (**e**) the variation curve of the TYDEX longitudinal slip ratio of the rear tire; (**d**) the variation curve of the radius.

In non-limit conditions, if the vehicle moves in a circle with a constant velocity and the yaw velocity based on the hypothesis in [32], the radius of the steady-state cornering is described as:

$$R = \frac{v}{\gamma} = \left(1 - \frac{m}{2\left(l_f + l_r\right)^2} \frac{l_f C_{f0} - l_r C_{r0}}{C_{f0} C_{r0}} v^2\right) \frac{l_f + l_r}{\delta_f} \tag{35}$$

where $C_{f0}$ and $C_{r0}$ are the cornering stiffnesses of front and rear tires respectively, when slip angles equal zero.

According to Equation (35), it suggests that the radius of the steady-state is smaller with the larger steering angle and the smaller velocity in non-limit conditions, which means the radius reaches the maximum value 3.67 m when the velocity is approaching to zero. Based on the comparison between the above analysis and Figure 4d, it suggests that vehicle in drift conditions can turn smaller corners and turn same corners with higher velocities than in non-limit conditions, which means the autonomous vehicle will take the full advantage of kinematic capability and be safer if it can realize drift motions.

### 4.2. Simulation Result

The controller is verified by simulation experiments in MATLAB/Simulink to achieve target circle motions in drift conditions and parameter values of target motions are shown in Table 2 including a circle motion with a maximum velocity.

The origin motions of Groups (a) and (b) both are the same uniform linear motion in non-limit motion as shown in the following, and the target motions are the drift circle motions as shown in Table 2a,b.

$$v = 8\text{ m/s}, \ \beta = \gamma = 0, \ \delta_f = 0, \ \kappa_f = \kappa_r = 0$$

Considering variation ranges of the steering angle and longitudinal slip ratios in practice, amplitudes and gains of them are restricted in simulations. The controller is used to achieve drift circle motions with uncertain disturbances and simulation results are shown in Figures 5 and 6 which respectively correspond to Groups (a) and (b) in Table 2, respectively. The uncertain disturbance $d_e$ is composed of three band-limited white noises which all observe the normal distribution and are same in the two simulations.

The conclusion is obtained by analyzing simulation results. The results in Figures 5a and 6a show that drift motions with different velocities can be realized whether the target velocity is the same as the origin or not, maximum velocity errors in control can be less than 10 percent of the target without regard to differences between the origin and the target velocities, and final velocities are stable near target velocities. As shown in Figures 5b and 6b, the controller can realize circle drift motion with both larger and smaller target sideslip angles. Figures 5c and 6c suggest that larger yaw rates in drift motions can be realized with stability. It can be seen from the Figures d and e of Figures 5 and 6 that steering angles and longitudinal slip ratios are stable with fine adjustments finally, but the system stability is unaffected because adjustments pale beside their values. The circle radiuses are calculated as the equation in Section 2.3 to show radius variations from the uniform linear origin motion to target drift circle motions in Figures 5f and 6f and include sudden changes because of lateral acceleration changes.

Combining all result figures, it is verified that the designed controller can accomplish the drift circle motions stably with robustness. Because of the neglect of slight differences during matching the tire model and target tire forces to obtain longitudinal slip ratios after gathering other targets, simulation results are not identical with targets and differences between results and targets of main status parameters, velocities, sideslip angles, yaw rates, and circle radiuses, are all less than 5 percent of targets, in spite of this, simulations are successful and finally stable near target stable groups in drift conditions and drift circle motions are realized. According to Figures 5 and 6, it can easily be found that the controller can effectively resist against uncertain external disturbances to realize drift motions.

### 5. Conclusions

This paper analyzes drift motion mechanisms in steady and transient states based on the theory-based method with the reference of the human operation by considering longitudinal, lateral, roll, and yaw motions and the rolling safety with the nonlinear tire model UniTire. It obtains the velocity limitation and main statue parameters including the side-slip angle and the yaw rate, and the drift motion characteristics are analyzed and

described. With the consideration of uncertain disturbances in practice, the state-feedback robust controller is designed based on LMIs and proposed to realize drift circle motions and to improve the driving safety of autonomous vehicles, and the robustness of the control system is verified by simulations in MATLAB/Simulink on the pavement whose friction coefficient equals 0.7 in the proposed study. The results show that vehicles that can drive in drift conditions are safer and the designed controller can realize drift circle motions as well as stability with robustness.

**Author Contributions:** Conceptualization, G.W. and D.X.; methodology, software, validation, and writing—original draft preparation, D.X.; writing—review and editing, supervision, and project administration, G.W.; investigation, D.X., L.Q. and C.G. All authors have read and agreed to the published version of the manuscript.

**Funding:** This research was funded by National Natural Science Foundation of China, Grant Number 51775548.

**Institutional Review Board Statement:** Not applicable.

**Informed Consent Statement:** Not applicable.

**Data Availability Statement:** Data is contained within the article. All data in this study can be obtained by calculation as shown.

**Acknowledgments:** The author(s) would like to thank to all the students taking part in the experiment from College of Engineering, China Agricultural University. D.X. and G.W. conceived motion mechanism analysis, control algorithm, performed the simulations, and finished the manuscript. L.Q. and C.G. revised the paper. And the authors gratefully acknowledge the financial support from the National Natural Science Foundation of China (No. 51775548).

**Conflicts of Interest:** The authors declare no conflict of interest.

## Abbreviations

The following abbreviations are basic parameters that is used in this manuscript:

| | |
|---|---|
| DOF | degree of freedom |
| LQR | linear quadratic regulator |
| index $i = f, r$ | to denote the front and rear axle respectively |
| $R$ | trajectory radius around mass center |
| $m$ | vehicle mass |
| $m_b$ | vehicle body mass (sprung mass) |
| $\delta_f$ | front-wheel steering angle |
| $v$ | vehicle velocity |
| $v_x$ | vehicle longitudinal velocity |
| $v_y$ | vehicle lateral velocity |
| $\beta$ | vehicle sideslip angle |
| $\gamma$ | vehicle yaw rate |
| $\phi$ | vehicle roll angle |
| $l_i$ | distance from gravity center to front or rear axle |
| $h_b$ | height of gravity center from the roll axis |
| $h_g$ | height of gravity center |
| $F_d$ | aerodynamic drag force |
| $\rho_a$ | air density |
| $C_d$ | aerodynamic drag coefficient |
| $A_f$ | frontal area of vehicle |
| $C_\phi$ | combined roll damping coefficient |
| $K_\phi$ | combined roll stiffness coefficient |
| $I_x$ | moment of inertia with respect to roll axis |
| $I_z$ | moment of inertia with respect to yaw axis |
| $I_{xz}$ | moment of inertia with respect to roll and yaw axis |
| $I_{x\phi}$ | moment of inertia with respect to roll axis after wheel lift-off |

| | |
|---|---|
| $I_{xz\phi}$ | moment of inertia with respect to roll and yaw axis after wheel lift-off |
| $\omega_i$ | wheel rotation angular velocity |
| $r_{ei}$ | wheel effective rolling radius |
| $\alpha_i$ | slip angle at each wheel |
| $\phi_i$ | normalized combined slip ratio at each tire |
| $\mu_i$ | friction coefficient between tire and road surface |
| $\overline{F}_i$ | normalized resultant force at the tire |
| $S_{ki}\ (k = x, y)$ | longitudinal or lateral slip ratio at each tire |
| $v_{ki}\ (k = x, y)$ | longitudinal or lateral velocity of the wheel center |
| $K_{ki}\ (k = x, y)$ | longitudinal slip or cornering stiffness of the tire respectively |
| $\phi_{ki}\ (k = x, y)$ | normalized longitudinal or lateral slip ratio at each tire |
| $\mu_{ki}\ (k = x, y)$ | longitudinal or lateral friction coefficient between tire and road surface |
| $\overline{F}_{ki}\ (k = x, y)$ | normalized longitudinal or lateral force at each tire |
| $F_{ki}\ (k = x, y, z)$ | longitudinal, lateral or vertical force at each tire |

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
