# Peer review of "Robust Control with Uncertain Disturbances for Vehicle Drift Motions"

_applsci, doi:10.3390/app11114917_

Round 1
Reviewer 1 Report
The paper is interesting. It concerns the analysis of the possibility of driving a car in drifting conditions in terms of driving safety by autonomous cars. Assessing the content of the paper, I think that it fully fits into the journal's profile.
The structure of the article is clear. A review of the knowledge on vehicle control with regard to motion in drift conditions is presented. On this basis, the need for in-depth research was formulated. The applied model of the car dynamics with 4 degrees of freedom, the so-called bicycle model with car body rolling motion. The non-linear UniTire model was used to describe the tangential forces in the tire-road contact. A robust control model (in which a linearized tire model was used) is described. The results of exemplary calculations for the assumed passenger car parameters and tire friction data and their analysis (in circular steady-state motion conditions and in the transient state of reaching the steady-state using the proposed controller) are presented. The conclusions indicated that promising results were obtained on the possibility of driving a vehicle in drifting conditions.
The ability to steer the vehicle in the states of high sideslip angles is attractive from the point of view of the vehicle's dynamic capabilities, its handling as well as overall safety. Autonomous vehicles open up new possibilities here. Theoretical research is of great value here and in this sense the article is valuable. In further work on this subject, it should also be considered what impact on the results will be the control of tangential forces through the operation of the actual drive system, as well as the forces of rolling resistance of tires and other excitations and disturbances.
Below are some comments that may improve the readability of the work or indicate minor faults.
General: It is mentioned that the applied UniTire model is a non-steady state model. However, in the presented description, I do not see a description of the transition states. Hence the question whether such transient tire states (lateral force relaxation) are actually included in the research.
- Line 12: “planer models” ? probably supposed to be "planar models"
- Line 76 – possibly an incorrectly cited reference number, [14] instead of [16]?
- line 107 – “origin of both coordinate” - One is oxyz and the other? Where is inertial coordinate system (Fig. 1)
- Eq. (1) – Please review again if the equations are correctly written (γ or dγ/dt?). Symbols K∅, C∅, not defined (in the section “abbreviations”)
- Eq. (5) -. Symbols IxF, Ixz∅, not defined (in the section “abbreviations”)
- Line 139 - What does the term "critical roll angle" mean ∅s?
- Eq. 4 – What is the physical meaning of "Vmin" and "Vmax"
- Line 339: In the section “Abbreviations” all abbreviations used in the text of the article are not explained (PILCO, MPC, NN, SDRE, MIMO, LPV, LMI, LQH ... etc)
- Line 382: item [16] is not quoted in the text of the work (see also remark to line 76)
Author Response
Thank you very much for your important suggestion. We have added this information as your suggestion.
Point 1:It is mentioned that the applied UniTire model is a non-steady state model. However, in the presented description, I do not see a description of the transition states. Hence the question whether such transient tire states (lateral force relaxation) are actually included in the research.
Response 1: UniTire model is a nonlinear and non-steady-state tire model and proposed by Guo Konghui who is a Chinese academician, which is more suitable to study vehicle drift motions, so there is no specific description in the abstract but the specific description is in Section 2.1.2. In this paper, we only display some inferences based on the tire model to simplify the whole derivation of the drift motion mechanism and don’t show the whole equations of the tire model. And in practice, we use the whole UniTire equations in the simulation model to sufficiently describe tire characteristics and linearize the tire model in the control system to simplify the controller state-space expressions. With the consideration of the understanding, the above has been added to this paper.
Point 2:Line 12: “planer models” ? probably supposed to be "planar models"
Response 2: Thank you very much for your important suggestion. The term “planer models” is a clerical error and has been changed to “planar models”.
Point 3:Line 76 – possibly an incorrectly cited reference number, [14] instead of [16]? Line 382: item [16] is not quoted in the text of the work (see also remark to line 76)
Response 3: Thank you very much for your important suggestion. The reference number has been corrected in Line 76 in the original paper.
Point 4:line 107 – “origin of both coordinate” - One is oxyz and the other? Where is inertial coordinate system (Fig. 1)
Response 4: Thank you very much for your important suggestion. The o in Line 107 in the original paper is the origin of the coordinate in Fig.1, and the description has been changed.
Point 5:Eq. (1) – Please review again if the equations are correctly written (γ or dγ/dt?).
Response 5: Thank you very much for your important suggestion. The clerical error in the third equation of Eq. (1) and the relevant copy errors have been corrected. These errors do not affect the simulations. Please see the attachment for equations.
Point 6:Eq. (1) –Symbols K∅, C∅, not defined (in the section “abbreviations”). Eq. (5) -. Symbols IxF, Ixz∅, not defined (in the section “abbreviations”)
Response 6: Thank you very much for your important suggestion. The undefined symbols in Eq. (1) and (5) have been added to the section “abbreviations”. Please see the attachment for details.
Point 7:Line 139 - What does the term "critical roll angle" mean ∅s?
Response 7: The “critical roll angle” in Line 139 in the original paper means the safe critical value while it’s dangerous after wheel lift-off. And the descriptions of these critical parameters have been changed to make them easier to understand. “Apparently, the relational expression between the safe roll angle , the safe roll rate and the safe roll angular acceleration in critical states can be obtained.”
Point 8:Eq. 4 – What is the physical meaning of "Vmin" and "Vmax"
Response 8: In Fig. 4(d), the symbols “vmin” and “vmax” describe the maximum and minimum permissible limit of the vehicle velocity in steady-state drifting motions, and the descriptions have been changed to more easily understand.
Point 9:Line 339: In the section “Abbreviations” all abbreviations used in the text of the article are not explained (PILCO, MPC, NN, SDRE, MIMO, LPV, LMI, LQH ... etc).
Response 9: Thank you very much for your important suggestion. The abbreviations, including PILCO, are explained in parentheses in section 1. The term “PILCO” means “Probabilistic Inference for Learning Control”. The term “MPC” means “Model Predictive Controller”. The term “NNs” means “Neural Networks”. The term “LQR” means “Linear Quadratic Regulator”. The term “LQ” means “Linear Quadratic”. The term “LMI” means “Linear Matrix Inequation”. The term “LPV” means “Linear Parameter Varying”. The term “T-S fuzzy model” means “Takagi-Sugeno fuzzy model”.

Reviewer 2 Report
- The abstract is weak for a few reasons.
- The very first sentence asserts two undefined terms (professor drivers and drift motions) that are likely to dissuade all readers from continuing unless the reader is already in the niched field that makes them already aware of the terms. After reading the entire manuscript, the reviewer remains uncertain the exact definitions intended by the authors, despite “drift motion” having been used twenty-nine times, albeit line 93 comes closest.
- While planer models is an obscure terms, the authors have elaborated sufficiently (the term implies neglecting rollover accident risk).
- Unitire is an undefined term that carries no meaning and negates the sentence containing its use.
- In lines18, robust theory is generally phrased, but LMI is an unknown acronym that is not defined for the reader.
- Realization of circle motion in drift conditions seems very odd (cars rarely ever seek to drive in circles), but perhaps the authors’ meaning has already been lost and the clarity of this sentence lies in proper understanding of the undefined terminology presented prior to line 18 and 19.
- The final sentence implies controllability to perform (presumed general) drift motions, but the previous sentences seem to imply the reader should only expect illustration of controllability when performing rare circular motions.
- Great job defining drifts and sharp turns and at least distinguishing professor drivers from ordinary drivers, although the assumed definitions of both terms would help.
- The literature review is modestly well done with only a few instances of abusing multiple citation without explanation. Each reference is briefly well-described with rationales for the reader to seek the cited reference.
- The claim of significance in line 85 should be eliminated. Firstly, significance will be established by the believable (repeatable) results achieved by the proposed methods. Secondly, referring to the literature review as “analysis” capable of validating claims of significance is disingenuous and detracts from the manuscript’s creditability. Lastly, the literature review neglects recently published development of deterministic artificial intelligence for unmanned vehicles [Sands, T. Development of deterministic artificial intelligence for unmanned underwater vehicles (UUV). Mar. Sci. Eng. 2020, 8(8),578.] in favor of classical methods included in the literature review: LQR, MPC, Hinf, fuzzy control, etc. and the omission detracts from claims of significance based on a select review of the literature.
- PILCO is undefined in line 33.
- MPC is undefined in line 36.
- NN is undefined in line 36.
- LQR is undefined in line 65.
- SDRE is undefined in line 66.
- MIMO is undefined in line 67.
- LQ, LMI, and LPV are undefined in line 75.
- T-S fuzzy is undefined in line 80
- UniTire is defined in line 99, but otherwise not articulated in brief verbiage telling the reader why they should care.
- Figures are well done (albeit a bit blurry).
- Figure 4 needs significant improvement to ensure the reader can discern the values on both abscissa(s), particularly in subplot (b) and (c).
- Identical line size/thickness and style (dotted/dashed/solid) making subplot (d) indistinguishable when the manuscript is printed, especially in black and white. The data lines must be modified to be sure the reader can distinguish the difference.
- Tables are well done and add value to the manuscript.
- Conclusions are poor due to several facets: brevity accompanying lack of understandable quantities results expressed in broadest terminology (identically absent from abstract). “Satisfying performance is validated” is a very weak conclusion, especially when so much quantitative data is immediately available (presented in the manuscript in qualitative plots). It seems that each and every plot of data provided in the manuscript could easily be accompanied by a table of data results (e.g. means and standard deviations of plots) and the results lead to a table of percent improvement after declaration of a benchmark case for comparison (all seemingly already available in the plots).
- The significance claim of line 85 would be easily illustrated with such results in the conclusions (e.g. xxx% performance improvement validates claims of significance of the proposed methods).
Author Response
Thank you very much for your important suggestion. We have added this information as your suggestion. And English language and style have been checked.
Point 1:The very first sentence asserts two undefined terms (professor drivers and drift motions) that are likely to dissuade all readers from continuing unless the reader is already in the niched field that makes them already aware of the terms. After reading the entire manuscript, the reviewer remains uncertain the exact definitions intended by the authors, despite “drift motion” having been used twenty-nine times, albeit line 93 comes closest.
Response 1: Thank you very much for your important suggestion. The terms “professor drivers” and “drift motions” have brief explanations. Professor drivers mean the highly competent drivers including racing drivers, who can drive cars in drift motions and the ordinary can’t. Drift motions are normally present in professional performances and car races such as the Formula 1 World Championship and are not considered in the autonomous drive. The explanation of “drift motions” is brief in the abstract based on the performance characteristics, and there is the other explanation in Section 1. Drift motions are realized by taking full advantage of tire forces against the ground when tires are in high slip ratios and the steering motion happens at the same time, which means tire forces reach the maximum and is very dangerous for ordinary drivers but can ensure maximum safety.
Point 2:While planer models is an obscure terms, the authors have elaborated sufficiently (the term implies neglecting rollover accident risk).
Response 2: The term “planer models” is a clerical error and has been changed to “planar models”.
Point 3:Unitire is an undefined term that carries no meaning and negates the sentence containing its use. UniTire is defined in line 99, but otherwise not articulated in brief verbiage telling the reader why they should care.
Response 3: UniTire model is a nonlinear and non-steady-state tire model and proposed by Guo Konghui who is a Chinese academician, which is more suitable to study vehicle drift motions, so there is no specific description in the abstract but the specific description is in Section 2.1.2. In this paper, we only display some inferences based on the tire model to simplify the whole derivation of the drift motion mechanism and don’t show the whole equations of the tire model. And in practice, we use the whole UniTire equations in the simulation model to sufficiently describe tire characteristics and linearize the tire model in the control system to simplify the controller state-space expressions. With the consideration of the understanding, the above has been added to this paper.
Point 4:In lines18, robust theory is generally phrased, but LMI is an unknown acronym that is not defined for the reader.
Response 4: The term “LMI” means “Linear Matrix Inequation” and is one way to solve the robust controller. The explanation is in the parenthesis after the term.
Point 5:Realization of circle motion in drift conditions seems very odd (cars rarely ever seek to drive in circles), but perhaps the authors’ meaning has already been lost and the clarity of this sentence lies in proper understanding of the undefined terminology presented prior to line 18 and 19.
Response 5: Normally, the vehicle rarely completes the whole circular motion and moves in a part of a circle to realize the turning. However, in drift motions, the drift circle motion is a classical motion, which is extensive in professional performances. And the study of the drift circle motion is the first and important step of the vehicle drift study. There is no explanation in the abstract and a brief explanation in Section 1.
Point 6:The claim of significance in line 85 should be eliminated. Firstly, significance will be established by the believable (repeatable) results achieved by the proposed methods. Secondly, referring to the literature review as “analysis” capable of validating claims of significance is disingenuous and detracts from the manuscript’s creditability. Lastly, the literature review neglects recently published development of deterministic artificial intelligence for unmanned vehicles [Sands, T. Development of deterministic artificial intelligence for unmanned underwater vehicles (UUV). Mar. Sci. Eng. 2020, 8(8),578.] in favor of classical methods included in the literature review: LQR, MPC, Hinf, fuzzy control, etc. and the omission detracts from claims of significance based on a select review of the literature.
Response 6: Thank you very much for your important suggestion. The claim of significance in Line 85 in the original paper has been eliminated as suggested. We have quoted “Artificial intelligence is most often expressed in stochastic algorithms that often have no knowledge whatsoever of the underlying problem being learned (a considerable strength of the methods)” in the paper “Development of deterministic artificial intelligence for unmanned vehicles (UUV)” as suggested to enrich the introduction.
Point 7:The abbreviations, PILCO, MPC, NN, LQR, ADRE, MIMO, LQ, LMI, LPV and T-S fuzzy, are undefined.
Response 7: Thank you very much for your important suggestion. The abbreviations, including PILCO, are explained in parentheses in section 1. The term “PILCO” means “Probabilistic Inference for Learning Control”. The term “MPC” means “Model Predictive Controller”. The term “NNs” means “Neural Networks”. The term “LQR” means “Linear Quadratic Regulator”. The term “LQ” means “Linear Quadratic”. The term “LMI” means “Linear Matrix Inequation”. The term “LPV” means “Linear Parameter Varying”. The term “T-S fuzzy model” means “Takagi-Sugeno fuzzy model”.
Point 8:Figures are well done (albeit a bit blurry). Figure 4 needs significant improvement to ensure the reader can discern the values on both abscissa(s), particularly in subplot (b) and (c). Identical line size/thickness and style (dotted/dashed/solid) making subplot (d) indistinguishable when the manuscript is printed, especially in black and white. The data lines must be modified to be sure the reader can distinguish the difference. Please see the attachment for figures.
Response 8: Thank you very much for your important suggestion. The original three-dimensional diagrams in Fig. 4 had grid lines which made figures blurry and the grid lines have been removed in the current three-dimensional diagrams in Fig. 4. The display angles of Fig. 4 (b) and (c) have been changed to ensure coordinate values clearness. The line type of vmin in Fig. 4 (d) has been changed to ensure the figure clearness when the manuscript is printed.
Point 9:Conclusions are poor due to several facets: brevity accompanying lack of understandable quantities results expressed in broadest terminology (identically absent from abstract). “Satisfying performance is validated” is a very weak conclusion, especially when so much quantitative data is immediately available (presented in the manuscript in qualitative plots). It seems that each and every plot of data provided in the manuscript could easily be accompanied by a table of data results (e.g. means and standard deviations of plots) and the results lead to a table of percent improvement after declaration of a benchmark case for comparison (all seemingly already available in the plots). The significance claim of line 85 would be easily illustrated with such results in the conclusions.
Response 9: Thank you very much for your important suggestion. Section 5 Conclusions has fine changes as suggested. “This paper analyzes drift motion mechanisms in steady and transient states based on the theory-based method with the reference of the human operation by considering longitudinal, lateral, roll, and yaw motions and the rolling safety with the nonlinear tire model UniTire and obtains the velocity limitation and main statue parameters including the side-slip angle and the yaw rate, and the drift motion characteristics are analyzed and described. With the consideration of uncertain disturbances in practice, the feedback robust controller is designed based on LMIs and proposed to realize drift circle motions and to improve the driving safety of autonomous vehicles, and the robustness of the control system is verified by simulations in MATLAB/Simulink on the pavement whose friction coefficient equals 0.7 in the proposed study. The results show that vehicles that can drive in drift conditions are safer and the designed controller can realize drift circle motions as well as stability with robustness.”

Round 2
Reviewer 2 Report
Thank you for the many accommodations of Reviewer recommendations.